# Piezoceramics Actuator with Attached Mass for Active Vibration Diagnostics of Reinforced Concrete Structures

**DOI:** 10.3390/s24072181

**Published:** 2024-03-28

**Authors:** Igor Shardakov, Aleksey Shestakov, Irina Glot, Georgii Gusev, Valery Epin, Roman Tsvetkov

**Affiliations:** Institute of Continuous Media Mechanics, Ural Branch of Russian Academy of Science, Academician Korolev Street 1, 614013 Perm, Russia; shap@icmm.ru (A.S.); glot@icmm.ru (I.G.); gusev.g@icmm.ru (G.G.); epin.v@icmm.ru (V.E.); flower@icmm.ru (R.T.)

**Keywords:** non-destructive testing, vibration diagnostics, piezoceramic actuator, attached mass, numerical simulation, electroelasticity

## Abstract

One of the effective methods of non-destructive testing of structures is active vibration diagnostics. This approach consists of the local dynamic impact of the actuator on the structure and the registration of the vibration response. Testing of massive reinforced concrete structures is carried out with the use of actuators, which are able to create sufficiently high-impact loads. The actuators, which are based on piezoelectric elements, cannot provide a sufficient level of force and the areas where it is possible to register the vibrations excited by such actuators are quite small. In this paper, we propose a variant of a piezoactuator with attached mass, which ensures an increase in the level of dynamic impact on the structure. The effectiveness of this version is verified by numerical modeling of the dynamic interaction of the actuator with a concrete slab. The simulation was carried out within the framework of the theory of elasticity and coupled electroelasticity. An algorithm for selecting the value of the attached mass is described. It is shown that when vibrations are excited in a massive concrete slab, an actuator with an attached mass of 1.3 kg provides a 10,000-fold increase in the force compared to an actuator without attached mass. In the pulse mode, a 100-fold increase in force is achieved.

## 1. Introduction

During the operation of complex engineering structures, emergency situations may arise due to the unacceptable level of their deformation. As a rule, this level is determined by the transition from an elastic deformation to an inelastic one and the appearance of cracks of different sizes. The occurrence of an unacceptable deformation or the assessment of the possibility of its occurrence can be revealed by means of structural health monitoring (SHM). One of the basic methods used in monitoring the stress–strain state of different engineering structures is vibration diagnostics. According to these methods, the state of a structure is estimated based on the analysis of its dynamic behavior. The application of different versions of this method is discussed in works [1,2,3,4,5,6,7].

Vibration diagnostics can be based on the registration of vibrations of a structure caused by random external disturbances, such as gusts of wind, the operation of technological equipment, passage of vehicles, microseismic vibrations of the earth’s surface, etc. Such an approach to vibration diagnostics is passive—it does not use special vibration sources. Its main disadvantage is that the disturbance regimes do not have the necessary repeatability. In addition, it may not be able to provide the required level of vibrations and set the required frequency range. An alternative approach is active vibration diagnostics. In this case, the necessary diagnostic vibration mode in the structure is created by means of special devices—actuators. They are distributed among structural elements and the diagnostic testing of the structure can include harmonic [8,9,10] or impulse [11,12] excitation of vibrations. The response of the structure to the diagnostic impact of the actuator is reflected in a set of vibrograms of spatially distributed response recording points. This set of vibrograms makes it possible to obtain information about the vibration profile of the structure at the time of implementation of the diagnostic impact. By repeating the diagnostic impacts with a certain periodicity, it is possible to obtain information on the vibrational behavior of the structure during its operation.

Analysis of the changes in the vibration profile allows us to establish the time at which the first signs of inelastic deformation appear in the structure, to find the location of inelastic deformation, to trace the process of crack growth, etc. The effectiveness in addressing these problems largely depends on the parameters characterizing the diagnostic mode produced by the actuator. First of all, these are the maximum level of mechanical impact on the structure and the required frequency range, in which this level of impact can be reached. The magnitude of the maximum mechanical impact determines the size of the process area. The required frequency range accounts for the excitation of natural vibration modes in the structure, which are most informative from the viewpoint of the appearance of zones of inelastic deformation, fracture, and crack growth. It should be noted that after reaching the maximum level of mechanical impact, the actuator itself should not break down and the structure should not go into the state of inelastic deformation.

Monitoring of large-scale building structures using active vibration diagnostics has a number of distinguishing features. In this case, the diagnostic impact on the structure must produce a sufficiently large force in a given frequency range. Only in this case it is possible to excite vibrations of the required natural modes and record the response of the structure at a sufficient distance from the point of impact. For this purpose, different electro-mechanical actuators are used [13]. However, they are rather heavy (their weight can be tens of kilograms) and automation of the control of such devices is quite problematic. Actuators based on piezoceramics seem to be more promising for use in online monitoring systems [9,14,15]. They offer a number of advantages. As a rule, they have no rubbing or impact-interacting parts, which ensures high reliability and accuracy of signal reproduction. Such actuators are able to produce impulse and harmonic excitations, the frequency of which can vary during diagnostic procedures. In addition, due to the use of a piezoelectric element, they are easily controlled and can be logically integrated into a system for automated monitoring of the deformation state of structures. Currently, there is a wide variety of actuators based on piezoceramics, the description of which can be found, for example, in the review [16]. However, most of them are devices designed to solve the problems of precise positioning of structural elements or ensure the functioning of different engines [17]. They are widely used in robotics. However, direct use of these devices for automated monitoring of large-scale structures is impossible. 

Currently, the most widely used approach is the embedding of a piezoceramic plate into the body of the structure to excite vibrations in it [8,10,18]. In this case, the piezoelectric element often acts as a vibration generator (actuator) and a vibration response recorder. The disadvantage of such a device is the inability to provide a sufficiently high level of force on the structure. In addition, rapid attenuation of the vibration process in concrete accounts for the fact that the area where registration of the vibration response is possible is rather small. There are also some approaches in which a piezoceramic plate is placed on the surface of the structure. The deformation state of structures is often assessed by the impedance method [19] but this approach also raises the question of the small size of the area, over which the signal generated by the piezoelectric element can propagate. Thus, the automated monitoring of massive reinforced concrete structures in the active vibration mode needs, on the one hand, the application of an actuator, which is based on piezoceramics, combining compactness and controllability, and on the other hand, a high level of mechanical impact on the structure in a given frequency range.

In this work, to solve these problems, we propose to use an actuator with an attached mass. A specific feature of this actuator is that the piezoceramic plate is complemented by a massive metal element and an adapter that allows the actuator to be located on the surface of the structure being inspected.

The study is carried out using the results of simulation of the interaction between the dynamic deformation of the actuator and a reinforced concrete slab. The simulation is carried out based on the fundamental principles of electroelasticity. Based on the results of numerical experiments, the possibility of increasing the force of the mechanical action of such a drive on the surface of a massive reinforced concrete slab is confirmed. An algorithm for determining the value of the attached mass is proposed. It ensures the maximum force action of the actuator in a given frequency range. The possibility of fracture of the ceramic plate due to the actuator operation in pulsed and harmonic modes are studied. The proposed actuator design provides a significant reduction in the level of maximum stresses in piezoceramics. This is achieved by introducing elements of conical coupling of the piezoelectric element with the attached mass and adapter. This design allows the actuator to operate at a higher level of force loading without destroying the piezoelectric element.

The developed piezoceramic actuator with an attached mass provides the generation of significant diagnostic forces and, therefore, increases the size of the diagnostic area of the structure. This, in turn, broadens the prospects of using such actuators in vibration diagnostics of different large-scale structures since it allows one to significantly increase the distances between actuators and sensors.

## 2. Materials and Methods

The design of the actuator under consideration is shown in Figure 1. The actuator consists of three main elements: a piezoceramic plate, an attached mass, and an adapter. The piezoceramic element is the main part of the actuator; it generates mechanical vibrations when an electrical potential difference is applied to it. It is made of lead zirconate titanate piezoceramics, which is one of the most common and commercially available piezoelectric materials. The attached mass is a metal part that enhances the effect of the actuator on the structure. The adapter is a metal element for mating the surface of a piezoceramic element with the surface of the structure.

Dynamic deformation processes arising from the interaction of the actuator with the structure during active vibration diagnostics are studied using numerical simulation based on the fundamental principles of electroelasticity. The computational scheme of the problem is shown in Figure 1. The mathematical formulation of the problem is made in the *Ox*_1_*x*_2_*x*_3_ rectangular coordinate system presented in Figure 1b. The *x*_2_ axis is directed normally to the plane of the figure and is not displayed. The actuator elements, namely attached mass 1, pesoceramic plate 2, and adapter 3, have a cylindrical shape and are rigidly fixed to each other; *x*_3_ is their common axis of symmetry. The piezoceramic element has electrical polarization along the vertical axis *x*_3_. Surfaces *x*_3_ = 0 and *x*_3_ = *h*_2_ are electrodes, the first has a positive electric potential and the second is zero. Tested structure 4 is a cylindrical concrete slab with a thickness of 100 mm and a diameter of 4000 mm. Its dimensions correspond to the characteristic dimensions of reinforced concrete slabs used in building structures. The geometric parameters of the actuator and the slab are given in Table 1. A general view of the actuator mounted on the concrete slab is shown in Figure 1a.

When choosing the frequency range of the dynamic impact produced by the actuator, it is necessary to make some assumptions about the nature of defects in the concrete slab, which can be identified in the process of vibration diagnostics. The appearance of the integrity defect in the form of a crack or a softened area with a characteristic size *L_d_* comparable to the thickness of the slab *h*_4_ (*L_d_* = 100 ± 50 mm) causes changes in the natural modes of vibration with a characteristic wavelength of periodicity *L* ≤ 100 ± 50 mm. 

At the first stage, it is necessary to approximately estimate the frequency range, at which the natural vibration modes with a wavelength comparable to the thickness of the slab (*L* < *h*_4_) are realized. In the determined frequency range, the actuator is expected to exert maximum force on the plate normal to its surface. This requirement is met by selecting the value of the attached mass of the actuator. It is found based on the results of the numerical solution of the problem of the dynamic interaction of the actuator with the plate using the fundamental principles of electroelasticity. In this case, the frequency of the electric potential set up on the piezoelectric element is selected from the determined frequency range. The magnitude of the force generated by the actuator should not lead to a fracture of the piezoelectric element. In the examined “actuator—plate” system, the piezoelectric element has the lowest strength characteristics. In order to realize an appropriate distribution of stresses in piezoceramics, it is proposed to introduce conical elements into the actuator to connect the metal parts of the actuator (adapter and attached mass) with the piezoelectric element. The effectiveness of this design is confirmed by the results of numerical simulation of the dynamic interaction of the actuator with the plate under the influence of a pulsed electric potential, which in this case results in a limiting stress–strain state of the piezoelectric element.

In the formulated mathematical problem, the attached mass, adapter, and plate are considered homogeneous and isotropic. The deformation state of the attached mass and adapter is described by the relations of the linear theory of elasticity [20] and the relations used to describe the behavior of the concrete slab take into account dissipation due to internal friction forces [21]. Piezoceramics is a polycrystalline material with anisotropic electroelastic characteristics. The electrical polarization of the piezoceramic element is induced along the *x*_3_ axis. The relationship between the electric field and the deformation state determines the ability of the piezoelectric element to change its dimensions under the influence of an electric field and generate an electric field when the dimensions change. The deformation state of the piezoceramic is described in terms of coupled electroelasticity [22]. The mathematical formulation of the problem of the deformation interaction of the actuator with the slab includes the following relations:Equations of motion:
(1)$$\frac{\partial {\sigma_{ij}}^{\left(k\right)}}{\partial {x_{j}}^{\left(k\right)}}=\rho^{\left(k\right)}\frac{\partial^{2}{U_{i}}^{\left(k\right)}}{\partial t^{2}};\;\;x^{\left(k\right)}=\left({x_{1}}^{\left(k\right)},{x_{2}}^{\left(k\right)},{x_{3}}^{\left(k\right)}\right)\in V^{\left(k\right)},\;i,j=(\bar{1,3}),\;k=(\bar{1,4}),$$
where the superscript (*k*) denotes the belongingness of the quantity to the corresponding body in the calculation diagram (see Figure 1); *σ_ij_*^(*k*)^ and *U_i_*^(*k*)^ are the components of the stress tensor and displacement vector in the Cartesian coordinate system; *ρ*^(*k*)^ and *V*^(*k*)^ are the density and volume of the corresponding body; *t* is time; and **x**^(*k*)^ is the radius vector of any point;Electrostatic equations for the piezoceramic element:
(2)$$\begin{array}{c} \nabla \cdot D=0;\\ E=-\nabla \varphi ; \end{array}\;x\in V^{\left(2\right)},$$where **D** = (*D*_1_, *D*_2_, *D*_3_) is the electric induction vector; ∇ = (∂/∂*x*_1_, ∂/∂*x*_2_, ∂/∂*x*_3_) is the nabla operator; **E** = (*E*_1_, *E*_2_, and *E*_3_) is the electric field strength vector; *φ* is the electrical potential; and *V*^(2)^ is the volume of the piezoceramic element;Geometric relations for strain tensor components ε*_ij_*^(*k*)^:
(3)$${\varepsilon_{ij}}^{\left(k\right)}=\frac{1}{2}\left(\frac{\partial {U_{i}}^{\left(k\right)}}{\partial {x_{j}}^{\left(k\right)}}+\frac{\partial {U_{j}}^{\left(k\right)}}{\partial {x_{i}}^{\left(k\right)}}\right);\;x^{\left(k\right)}\in V^{\left(k\right)},\;k=\bar{1,4},\;i,j=\bar{1,3};$$Physical relations for piezoceramics polarized along the *x*_3_ axis (the superscript *k* = 2, which designates the belongingness to a piezoelectric element, is omitted here):
(4)σ11=c11ε11+c12ε12+c13ε33−e31E3;σ22=c12ε11+c11ε22+c13ε33−e31E3;σ33=c13ε11+c13ε22+c33ε33−e33E3;σ23=c44ε23−e15E2;σ13=c44ε13−e15E1;σ12=12c11−c12ε12;D1=α11E1+e15ε12;D2=α11E2+e15ε23;D3=α33E3+e31ε11+e31ε22+e33ε33;  x∈V2,where *c*_11_, *c*_12_, *c*_11_, *c*_13_, *c*_33_, and *c*_44_ are the elastic moduli; *e*_31_, *e*_15_, and *e*_33_ are the piezoelectric moduli; and *α*_11_ and *α*_33_ are the dielectric constant coefficients;Physical relations for metal
(5)$${\sigma_{ij}}^{\left(k\right)}=K^{\left(k\right)}\theta^{\left(k\right)}\delta_{ij}+2G^{\left(k\right)}{d_{ij}}^{\left(k\right)},\;\;x\in V^{\left(k\right)},\;\;k=(1,3);$$Physical relations for concrete
(6)$${\sigma_{ij}}^{\left(4\right)}=K^{\left(4\right)}\theta^{\left(4\right)}\delta_{ij}+2G^{\left(4\right)}{d_{ij}}^{\left(4\right)}+\beta \left[K{\dot{\theta }}^{\left(4\right)}\delta_{ij}+2G^{\left(4\right)}{{\dot{d}}_{ij}}^{\left(4\right)}\right],\;x\in V^{\left(4\right)}.$$

In relations (5) and (6), *d_ij_*^(*k*)^
*= ε_ij_*^(*k*)^
*−* 1/3∙*θ*^(*k*)^*δij* are the components of the deformation tensor deviator, *θ*^(*k*)^ = *ε*_11_^(*k*)^ + *ε*_22_^(*k*)^ + *ε*_33_^(*k*)^ is the volumetric deformation and the dot means the time derivative; *K*^(*k*)^ and *G*^(*k*)^ are the volumetric and shear moduli of the corresponding materials; *β* is the parameter that determines the dissipative properties of concrete [18]; *δ_ij_* is the Kronecker symbol; and $x\in V^{\left(k\right)},\;k=(1,3,4)$.

The boundary conditions for the «attached mass–piezoelectric element–adapter–slab» system are written as follows:(7)$$\begin{array}{l} \varphi =0,\;\;x\in S^{0},\\ \varphi =\varphi^{*},\;x\in S^{+}, \end{array}\;\begin{array}{l} {U_{i}}^{\left(k\right)}={U_{i}}^{\left(k+1\right)},\;{\sigma_{ij}}^{\left(k\right)}{n_{j}}^{\left(k\right)}={\sigma_{ij}}^{\left(k+1\right)}{n_{j}}^{\left(k+1\right)},\;x\in S^{\left(k,k+1\right)},\;k=\left(\bar{1,3}\right),\\ {\sigma_{ij}}^{\left(m\right)}{n_{j}}^{\left(m\right)}=0,\;x\in {S_{f}}^{\left(m\right)},\;{U_{i}}^{\left(m\right)}=0,\;\;x\in {S_{U}}^{\left(m\right)},\;m=\left(\bar{1,4}\right), \end{array}i,j=\left(\bar{1,3}\right),$$
where *S*^0^ and *S*^+^ are the electrode surfaces of a piezoceramic element with zero and specified electric potential *φ*^*^; *S*^(*k*,*k*+1)^ is the contact surface of the *k*-th and *k*+1-th bodies; *S_f_*^(*k*)^ and *S_U_*^(*k*)^ are the free and rigidly fixed surfaces of the *k*-th body; and *n_f_*^(*k*)^ are the components of the unit vector normal to the surface of the *k*-th body.

The calculations use the electrical and mechanical characteristics of the piezoceramics CTS-19 [22], given in Table 2. The parameters of concrete and metal are given in Table 3. The permissible value of the electric field strength applied to piezoceramics is 2 × 10^6^ V/m. The limiting stresses for piezoceramics are as follows: compression is 290 MPa and tension 15 MPa.

A solution to the problem of natural vibration modes and natural frequencies of the elastic concrete slab was obtained using relations (1), (5), and (6) of the mathematical formulation. Solving this problem allowed us to identify the natural modes, which are most sensitive to the appearance of integrity defects in the tested slab, the size of which is comparable to the thickness of the slab, as well as to identify the frequency range corresponding to these vibration modes.

The force of action of the actuator on the slab at different values of the attached mass was calculated by performing a series of calculations in the framework of the mathematical formulation of the coupled problem of electroelasticity (1)–(7). The response of the slab to the impact of the actuator in the form of harmonic and pulse excitation was analyzed. Based on the results of a numerical experiment, the dependence of the maximum actuator force on the value of the attached mass was obtained.

Analysis of the stress–strain state in the elements of the «attached mass–piezoelectric element–adapter–slab» system was performed under pulsed excitation of the piezoelectric element. As a result, the maximum values of tensile and compressive stresses in the piezoceramic element were determined and compared with the maximum acceptable values.

The numerical implementation of the problem was carried out by the finite element method using the ANSYS software tool (version 2022 R2). The solution was constructed based on the tetrahedral 10-node finite element with a quadratic approximation of displacements and a linear approximation of the electric potential. A non-uniform finite element mesh with concentration in the piezoceramics zone was used (Figure 2a). The rational size of the elements and required time steps were determined in the course of numerical experiments. The dependence of the stress *σ**_33_ in piezoceramics on the characteristic element size (Figure 2b) demonstrates the stabilization when the element size is less than 2.5 mm. When analyzing dynamic processes, a time step of 4 μs was used.

## 3. Results and Discussion

The results of solving the spectral problem are presented in Figure 3. The figure shows the spatial configuration of several eigenmodes and corresponding eigenfrequencies. In terms of the future prospects of vibration diagnostics, of particular interest is the spectrum of natural frequencies, which correspond to natural vibration modes with a periodicity wavelength comparable to the thickness of the slab (*L* ≤ 100 ± 50 mm). Numerical analysis showed that natural modes that meet this condition are located in the frequency range from 4 kHz to 6 kHz. These modes are bending vibrations, which are found to be most sensitive to the appearance of defects in the structure comparable in size to the thickness of the slab. It should be noted that in the process of vibration diagnostics, these eigenmodes and their corresponding frequencies respond to the defects in the form of transverse cracks, which are comparable in scale to the characteristic distance between the nodal lines of the eigenmodes.

These eigenmodes and frequencies are well registered by existing measuring devices since they contain significant kinematic quantities (displacements, speed, and acceleration) directed normally to the surface of the slab. It was assumed that the frequency range found in this way (4–6 kHz) is the working interval for the actuator and at these frequencies, the actuator should exert maximum force on the slab normal to its surface.

For the developed actuator, the effect of increasing force impact is achieved by choosing the value of the attached mass. The mass value was determined based on the results of the numerical simulation of the vibration process in the slab during harmonic excitation of piezoceramics. An electric potential *φ**(*t*) applied to the electrode surfaces varied harmonically in time with an amplitude of 1 V. The signal frequency varied in the range from 100 Hz to 6000 Hz. The value of the added mass was set in the range of 0 to 2 kg with an increment of 0.1 kg. The amplitude–frequency response (AFR) of the actuator impact force normal to the surface of the slab was determined for each specified value of the attached mass. The AFR, at which the actuator force was maximum at a frequency of about 5 kHz, was selected from the entire set of frequency responses. This frequency corresponds to the previously established operating range of the actuator. It was established that this frequency response was provided by an actuator with an attached mass of 1.3 kg. The AFR corresponding to this mass is shown in Figure 4a,b on the linear and logarithmic scales. For comparison, Figure 4c,d shows similar frequency responses obtained at zero added mass.

As it follows from the dependences in Figure 4a,b, the maximum force that can be generated by an actuator with an attached mass of 1.3 kg is 5.7 N. It is reached at a frequency of 5000 Hz. A comparison of the results in Figure 4a,b with the dependencies in Figure 4c,d, shows that the use of an attached mass makes it possible to increase the force by 10,000 times. Figure 5 presents the image of the deformed state of the slab and actuator under harmonic impact with a frequency of 4400 Hz. The forced vibration mode of the concrete slab at a certain instant of time is shown along with the field of the vertical acceleration component *a*_3_.

One of the diagnostic operating modes of the actuator is pulse loading, which triggers a deformation wave process in the structure. Below are the results of the numerical experiment simulating the response of a concrete slab to the pulse action of actuators with different attached masses (0.00 kg, 0.13 kg, 0.65 kg, and 1.3 kg). An electric potential pulse *φ**(*t*), shown in Figure 6a, is applied to the piezoceramic element of the actuator. The signal amplitude of 10 kV is selected based on the condition that in a given piezoceramic material, the limiting value of the electric field strength has been achieved. A pulse duration of 0.22 ms ensures the excitation of a wave process in the slab with a frequency of about 5 kHz.

Figure 6b shows the result of the simulation of the force acting on the concrete slab from the actuator with an attached mass of 1.3 kg. The maximum value of this force is 2.4 kN. The results of a numerical experiment simulating the wave process in the slab excited by an actuator with different attached masses were used to plot the maximum force versus the additional mass shown in Figure 7. It can be seen from the figure that in the case of zero added mass, this force is 34 N. Thus, an actuator with an attached mass of 1.3 kg makes it possible to increase the force on the slab by almost 100 times compared to an actuator that has no additional mass.

Figure 8 and Figure 9 demonstrate the deformation state of the actuator with an attached mass of 1.3 kg and the concrete slab under pulsed action. Figure 8 shows the field of vertical displacements at different times within the first 210 µs after starting pulse loading. Figure 9 shows the shape of the wave excited in the slab. The acceleration amplitude at the periphery of the slab reaches 24 m/s^2^ and this value can be measured by most modern accelerometers.

The results of the calculation of the stress–strain state in the elements of the «attached mass–piezoelectric element–adapter–slab» system in the pulse excitation mode allowed us to determine the maximum values of tensile and compressive stresses in the piezoceramic element and compare them with the maximum permissible values. It should be noted that piezoceramics have significantly lower values of the maximum permissible characteristics compared to the characteristics of the metal parts of the actuator and the concrete slab. The limiting tensile and compressive stresses for piezoceramics are 15 MPa and 290 MPa, respectively. Calculations have shown that a stress field with a dominant normal stress *σ*_33_ is formed in the piezoelectric element. 

In an actuator with an attached mass of 1.3 kg, the maximum tensile stress *σ*_33_ is realized in the vicinity of the points of contact of the piezoceramics with the attached mass and the piezoceramics with the adapter. With a maximum permissible electric potential of 10 kV amplitude, the maximum tensile stress is achieved at 41 MPa, which significantly exceeds the maximum permissible value for piezoceramics (15 MPa). 

To eliminate this undesirable situation, it is proposed to change the configuration of the interface between the piezoelectric element and the attached mass, as well as the piezoelectric element and the adapter. Figure 10 shows two variants of the contact zone configuration. Figure 10a shows a variant with conical transition zones and Figure 10b shows the option without a transition zone used in the above-mentioned computational schemes. All conical surfaces have a cone angle of 45°.

Figure 11 shows the fields of the normal stress component *σ*_33_ in the diametrical section of the piezoceramic element, corresponding to the two design schemes. These stress fields correspond to 0.13 ms after the beginning of the pulse action, the electric potential amplitude of 10 kV, and the additional mass of 1.3 kg.

A comparison of the results presented in Figure 11 demonstrates that the use of conical zones of the interface of the piezoceramic element with attached mass and the adapter can significantly reduce the maximum values of tensile stresses *σ*_33_. Compared to the design without transition elements, these values decreased by a factor of four (from 41 MPa to 10 MPa) and became less than the maximum permissible value of 15 MPa. Thus, the presence of conical transition zones ensures the integrity of the piezoceramic element under pulsed loading, whereas in a structure without such zones, the piezoceramics can break down.

## 4. Conclusions

In this work, we have considered a version of a piezoceramics actuator with attached mass. The actuator is designed to generate elastic waves in large-scale reinforced concrete structures. It provides excitation of vibrations in a given frequency and amplitude range. The main field of application of such devices is active vibration diagnostics of structures controlled by automated deformation monitoring systems. A specific feature of the proposed actuator is the presence of an attached mass, which ensures the required mode of dynamic action on the surface of the structure.

Numerical simulation of different modes of dynamic deformation interaction of the actuator with the surface of a massive concrete slab was carried out within the framework of the theory of elasticity and coupled electroelasticity. The frequency range in which the actuator could generate the steady-state vibration and wave processes in the slab has been established. 

The procedure for selecting the frequency range, in which the actuator is expected to produce a dynamic effect, has been described. It is shown that during vibration diagnostics of a massive reinforced concrete slab, the operating frequencies of the actuator should be in the range of 4 to 6 kHz. An algorithm for determining an additional mass that can significantly increase the action of the actuator has been proposed. It is based on a comparison of amplitude–frequency characteristics (maximum actuator force—frequency) obtained at different values of the added mass. It has been found that the attached mass of 1.3 kg provides the maximum mechanical impact of the actuator in the operating frequency range. Compared to an actuator without an attached mass, the developed version of the actuator provides the following effect:−A 10,000-fold increase in the force amplitude during harmonic excitation; −A 100-fold increase in the maximum force during pulse excitation.

The most stressed zones of the piezoelectric element have been determined by numerical modeling of the stress–strain state of the actuator in the process of pulse and harmonic impact on the concrete structure. It has been found that during pulse loading, the scheme without transition zones for coupling the piezoelectric element with other parts of the actuator leads to the appearance of tensile stresses of 41 MPa in the piezoceramics. This exceeds the permissible values of 15 MPa. The design with conical coupling of the piezoelectric element with additional mass and adapter makes it possible to reduce the maximum tensile stresses in the piezoceramics to 10 MPa, i.e., by four times. Thus, the actuator of this design ensures the integrity of the piezoceramics during operation.

The developed version of the device adds to the arsenal of actuators intended for use in automated monitoring systems to generate dynamic impacts on large-scale reinforced concrete structures. The proposed algorithm for determining the actuator parameters can be used in the development and creation of active vibration diagnostics systems for any large-scale engineering and building structures.

## Figures and Tables

**Figure 1 sensors-24-02181-f001:**
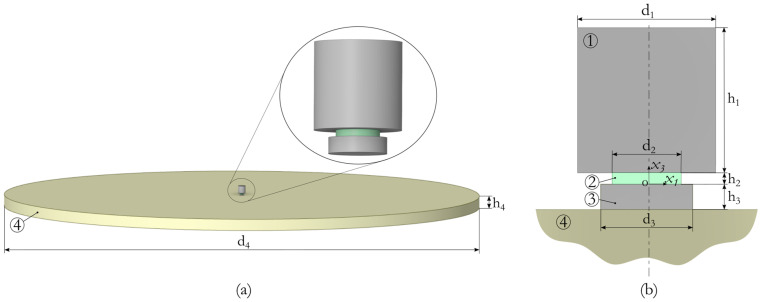
The design scheme of the actuator: (**a**) general view and (**b**) axial sections; 1—attached mass, 2—piezoceramic element, 3—adapter, and 4—concrete slab.

**Figure 2 sensors-24-02181-f002:**
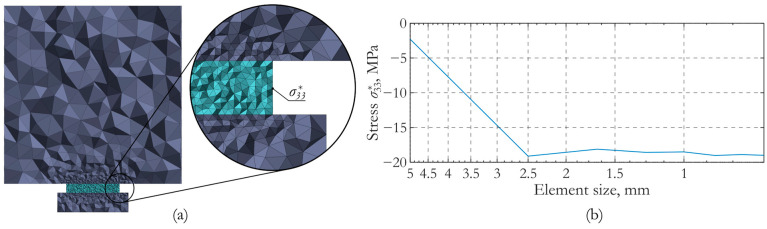
Details of the numerical implementation: (**a**) finite element approximation and (**b**) dependence of the stress component *σ**_33_ on the size of the finite elements in the piezoceramics.

**Figure 3 sensors-24-02181-f003:**
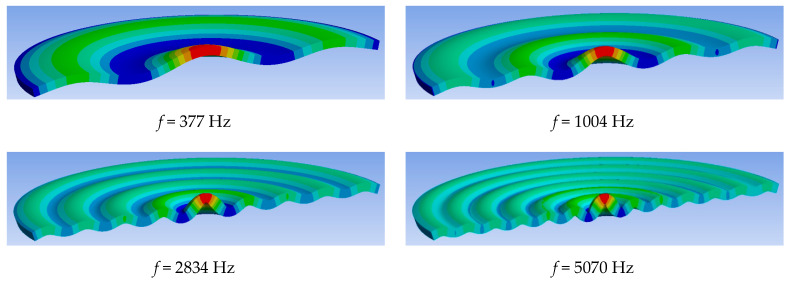
Spatial configurations of natural vibration modes of an elastic concrete slab with natural frequencies *f* from 377 Hz to 5 kHz.

**Figure 4 sensors-24-02181-f004:**
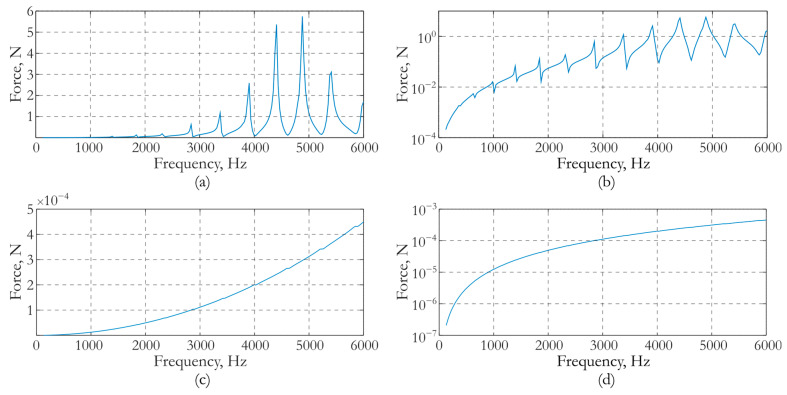
Amplitude–frequency response of the force caused by an actuator: (**a**,**b**) attached mass is 1.3 kg and (**c**,**d**) attached mass is zero.(**a**,**c**) Linear scale; (**b**,**d**) Logarithmic scale.

**Figure 5 sensors-24-02181-f005:**
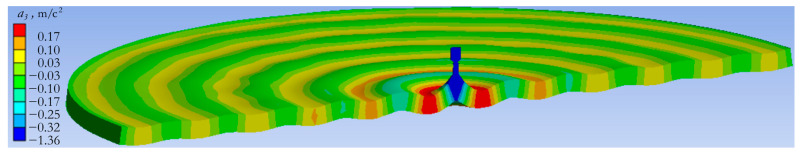
Forced vibration mode of the concrete slab under the harmonic excitation of an actuator with a frequency of 4400 Hz and an acceleration field at a fixed point in time.

**Figure 6 sensors-24-02181-f006:**
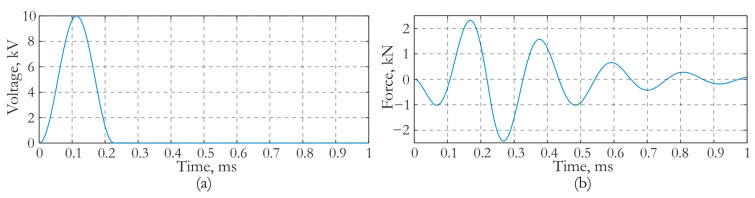
(**a**) The electrical potential difference applied to the piezoceramic element and (**b**) the force exerted by the actuator on the concrete slab.

**Figure 7 sensors-24-02181-f007:**
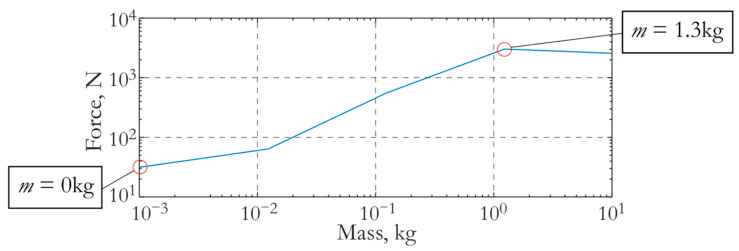
The maximum force of the actuator on the concrete slab depends on the attached mass.

**Figure 8 sensors-24-02181-f008:**
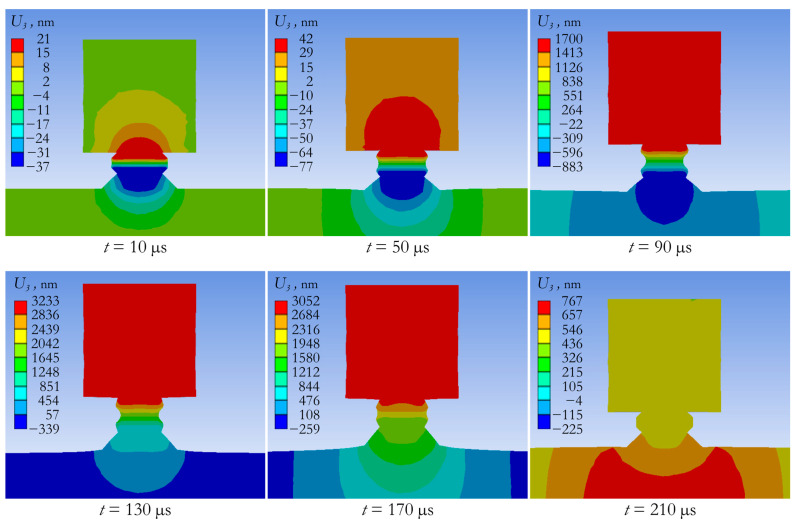
Field of vertical displacements in the actuator and slab at different time steps.

**Figure 9 sensors-24-02181-f009:**
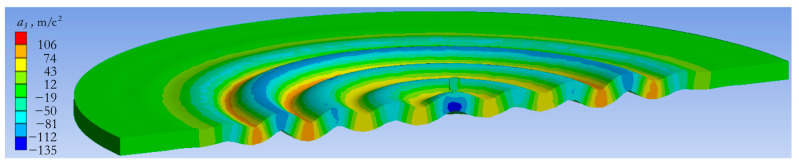
Response of the concrete slab to the impulse action of the actuator: field of displacements and accelerations at a fixed point in time.

**Figure 10 sensors-24-02181-f010:**
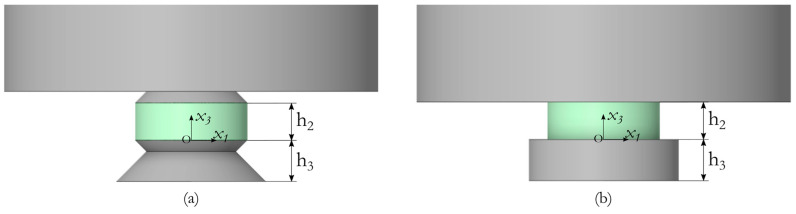
Design schemes: (**a**) Actuator with conical elements and (**b**) actuator without conical elements.

**Figure 11 sensors-24-02181-f011:**
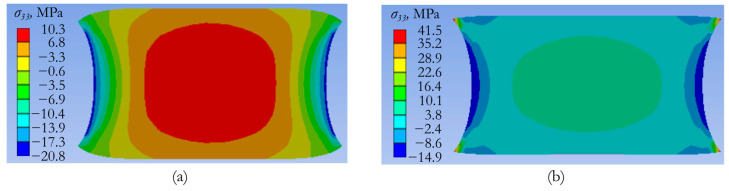
Isofields of the normal stress component *σ*_33_ in the diametrical section of the piezoceramic element: (**a**) design scheme with conical elements and (**b**) design scheme without conical elements.

**Table 1 sensors-24-02181-t001:** Geometric parameters of the design scheme.

d_1_, mm	h_1_, mm	d_2_, mm	h_2_, mm	d_3_, mm	h_3_, mm	d_4_, mm	h_4_, mm
60	60	30	5	40	11	4000	100

**Table 2 sensors-24-02181-t002:** Electromechanical parameters of piezoceramics.

*c*_11_ GPa	*c*_12_ GPa	*c*_13_ GPa	*c*_33_ GPa	*c*_44_ GPa	*ε*_11_/*ε*_0_	*ε*_33_/*ε*_0_	*ε*_0_ F/m	*d*_33_ K/N	*e*_31_ K/m^2^	*e*_33_ K/m^2^	*e*_15_ K/m^2^	*ρ* kg/m^3^
109	61	57	93	24	840	820	8.85 × 10^−12^	304 × 10^−12^	−4.9	14.9	10.9	7740

**Table 3 sensors-24-02181-t003:** Mechanical parameters of concrete and metal.

*K*^(1)^, *K*^(3)^ GPa	*G*^(1)^, *G*^(3)^ GPa	*ρ*^(1)^, *ρ*^(3)^ kg/m^3^	*K*^(4)^ GPa	*G*^(4)^ GPa	*ρ*^(4)^ kg/m^3^	*β* s
166.6	76.9	7850	26.8	14.5	2400	3464 × 10^−7^

## Data Availability

The data presented in this study are available on request from the corresponding author.

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
