# Peer review of "Piezoceramics Actuator with Attached Mass for Active Vibration Diagnostics of Reinforced Concrete Structures"

_sensors, 2024, doi:10.3390/s24072181_

Round 1

Reviewer 1 Report

Comments and Suggestions for Authors

Dear Authors,

The manuscript should be revised before it is published. My remarks and questions are listed in the attachment.

Kind Regards 

Comments on the Quality of English Language

English should be polished

Author Response

We thank the reviewer for his attention to our manuscript for his insightful and useful comments, which encouraged us to present our results more clearly and substantively.

You can find our response to reviewer 1’s comments in the attached file.

In the revised version of our manuscript, we have highlighted in purple the passages to which we have made significant changes or additions. We have highlighted additional items in the reference list in blue. We have also marked changes in figure numbers and reference numbers in blue. We have tried to improve our English, but we have not noted the text changes made for this purpose.

Reviewer 2 Report

Comments and Suggestions for Authors

This paper presents a method to increase the dynamic impact level of a piezoceramic actuator through the use of additional mass. The effectiveness of the device is verified by numerical simulations of the dynamic interaction of the actuator with a concrete slab. The simulations are carried out in the framework of the theory of elasticity and coupled electroelasticity. The obtained results make it possible to determine reasonable geometrical and mass parameters of the actuator. The research in this paper is of great engineering and academic significance, but the following problems still exist and it is suggested to revise it for further review.

1. In the introduction, it is recommended that the concept of the proposed new piezoelectric accessory actuator be described in more detail, including its design principle, working mechanism, and advantages and innovations over conventional piezoelectric elements. At the end of the introduction, it is recommended to briefly summarize the overall framework and main contributions of the study.

2. In describing the numerical simulation methodology used in the experiments, it is possible to explain in more detail why the finite element methodology and the ANSYS software tool are chosen, and it is possible to explain the basis for the choices made regarding the degree of discretization of the finite elements and the time steps required.

3. In discussing the experimental results, it is suggested to characterize in more detail the forces acting on the concrete slab for different values of additional mass and to elucidate the relationship between these forces and the response of the concrete slab.

4. In the final conclusion part, it is recommended that the main contributions of the research work be described more prominently and specifically.

5. There are some grammatical problems in this paper, and it is recommended that the full text be checked and revised.

Comments on the Quality of English Language

English editing is required

Author Response

We thank the reviewer for his attention to our manuscript for his insightful and useful comments, which encouraged us to present our results more clearly and substantively.

You can find our response to reviewer comments in the attached file.

In the revised version of our manuscript, we have highlighted in purple the passages to which we have made significant changes or additions. We have highlighted additional items in the reference list in blue. We have also marked changes in figure numbers and reference numbers in blue. We have tried to improve our English, but we have not noted the text changes made for this purpose.

Reviewer 3 Report

Comments and Suggestions for Authors

This paper introduces a method for increasing the level of dynamic impact of a piezoceramic actuator using an attached mass, which is an effective way to conduct active vibration diagnostics of reinforced concrete structures. The paper is well organized. I think it can be published after minor revision, and the advice for the revisions are listed below.

1.  The last paragraph of Introduction is not necessary, and it should be deleted.

2. The conclusion should be rewritten as a more concise version, and I recommend that the findings should be listed as general descriptions in number.

Author Response

We thank the reviewer for his attention to our manuscript for  useful comments, which encouraged us to present our results more clearly and substantively.

Our response to the reviewer's comments can be found in the attached file.

In the revised version of our manuscript, we have highlighted in purple the passages to which we have made significant changes or additions. We have highlighted additional items in the reference list in blue. We have also marked changes in figure numbers and reference numbers in blue. We have tried to improve our English, but we have not noted the text changes made for this purpose.

Reviewer 4 Report

Comments and Suggestions for Authors

This manuscript proposes a version of a piezoceramic actuator with an attached mass for Active Vibration Diagnostics of Reinforced Concrete Structures by increasing the magnitude.  For this end, firstly authors give the basic mathematical equations. And then, they give the numerical illustration. However, the following issues need to be considered:

1.      The innovation of this work is not clear. This is because this version is not new. Thus, authors should clearly describe the innovation of this work.

2.      The vibration equation must be derived.

3.      The parameter sensitivity should be considered. Meantime, the corresponding physical meaning should be given.

Author Response

We thank the reviewer for his attention to our manuscript for his  useful comments, which encouraged us to present our results more clearly and substantively

Our response to the reviewer's comments can be found in the attached file.

In the revised version of our manuscript, we have highlighted in purple the passages to which we have made significant changes or additions. We have highlighted additional items in the reference list in blue. We have also marked changes in figure numbers and reference numbers in blue. We have tried to improve our English, but we have not noted the text changes made for this purpose.

Round 2

Reviewer 1 Report

Comments and Suggestions for Authors

Dear Authors,

I have no further questions related to revised version of the manuscript. 

The manuscript can be published after minor revision (methodological errors and text editing).  

Kind Regards

Comments on the Quality of English Language

Quality of English is fine. 

Reviewer 4 Report

Comments and Suggestions for Authors

The revised manuscript has been improved according to the comments. Thus, it can be considered to be accepted for publication.